# Codesigning an intervention to strengthen COVID-19 vaccine uptake in Congolese migrants in the UK (LISOLO MALAMU): a participatory qualitative study protocol

Alison F Crawshaw [1], Caroline Hickey,[2] Laura Muzinga Lutumba,[3] Lusau Mimi Kitoko,[3] Sarah Nkembi,[3] Felicity Knights,[1] Yusuf Ciftci,[4] Lucy Pollyanna Goldsmith [5] Tushna Vandrevala,[6] Alice S Forster,[7] Sally Hargreaves [1]

For numbered affiliations see end of article.

**Correspondence to**
Dr Sally Hargreaves;
s.hargreaves@sgul.ac.uk

## ABSTRACT

**Introduction** Migrants positively contribute to host societies yet experience barriers to health and vaccination services and systems and are considered to be an underimmunised group in many European countries. The COVID-19 pandemic has highlighted stark inequities in vaccine uptake, with migrants facing access and informational barriers and lower vaccine confidence. A key challenge, therefore, is developing tailored vaccination interventions, services and systems which account for and respond to the unique drivers of vaccine uptake in different migrant populations. Participatory research approaches, which meaningfully involve communities in co-constructing knowledge and solutions, have generated considerable interest in recent years for those tasked with designing and delivering public health interventions. How such approaches can be used to strengthen initiatives for COVID-19 and routine vaccination merits greater consideration.

**Methods and analysis** LISOLO MALAMU ('Good Talk') is a community-based participatory research study which uses qualitative and coproduction methodologies to involve adult Congolese migrants in developing a tailored intervention to increase COVID-19 vaccine uptake. Led by a community–academic coalition, the study will involve (1) semistructured in-depth interviews with adult Congolese migrants (born in Democratic Republic of Congo, >18 years), (2) interviews with professional stakeholders and (3) codesign workshops with adult Congolese migrants. Qualitative data will be analysed collaboratively using reflexive thematic analysis, and behaviour change theory will be used in parallel to support the coproduction of interventions and make recommendations across socioecological levels. The study will run from approximately November 2021 to November 2022.

**Ethics and dissemination** Ethics approval was granted by the St George's University Research Ethics Committee (REC reference: 2021.0128). Study findings will be disseminated to a range of local, national and international audiences, and a community celebration event will be held to show impact and recognise contributions.

## STRENGTHS AND LIMITATIONS OF THIS STUDY

⇒ This study uses community-based participatory research approaches, which promote principles of inclusivity and power sharing. An academic–community partnership ('study coalition') including three members of the target Congolese population was formed to codesign and deliver the study and cowrote this protocol.

⇒ The main research topic of COVID-19 vaccination was driven by the desires and needs of the study population and interventions will be coproduced which are informed by lived experience, insider knowledge and perspectives.

⇒ Because the study coalition involves members of the target population who will act as peer researchers, recruitment and research activities will be designed and conducted in ways (times, locations, formats, etc) that are appropriate for the target population, and may therefore be more acceptable and foster increased levels of trust, which can increase validity and likelihood of success of community-led interventions.

⇒ Building trust between the local and wider community, stakeholders and academic partners was a continuous process which began prior to study conception.

⇒ St George's, University of London were lead recipients of the funding, which inherently skewed the power balance between partners, but efforts were made to overcome this, for example, by giving the Congolese-led community-based organisation control over the spending and use of funds and providing them with skills-based training in budget management.

Recommendations for implementation and evaluation of prototyped interventions will be made.

## INTRODUCTION

Migrants (defined here as foreign-born individuals, see box 1) contribute positively to their host societies[1] but many continue to

> **Box 1  A note on the challenges around existing definitions and language used to speak about migrants and other minoritised populations**
>
> There is no internationally agreed definition of 'migrant', but for the purpose of this protocol we have defined a migrant as a foreign-born individual. Much of the language used to talk about migrants (and other minoritised populations) in public health is influenced by extant literature, databases/national registers used in population health and international policies, many of which use inconsistent or inappropriate definitions and groupings of migrants, or fail to record migrants at all. This language and the limitations of existing data are problematic and incongruent with a community-centred approach which seeks to redistribute power, address injustices and decolonise medical practices. In our introduction we discuss recent literature regarding vaccination uptake in individuals grouped varyingly by race, ethnic group or migrant status, depending on the citing source, in order to provide context to the research problem. We recognise the limitations of these groupings and suggest higher standards will be essential in addressing the needs of diverse populations.

be excluded from health and vaccination services and systems worldwide, are considered to be an underimmunised group and suffer worse health outcomes than the general population.[1–5] This has been brought to light acutely during the COVID-19 pandemic, where migrants have been disproportionately represented in COVID-19 deaths and all-cause mortality,[2 6] although even before this migrants (particularly those from low and middle-income countries) were known to be at risk of underimmunisation for routine vaccinations[4 5 7 8] and involved in outbreaks of serious vaccine-preventable diseases, including measles.[9] In addition to these risks, many migrant and refugee populations have now been shown to be more reluctant to vaccinate for COVID-19 and to have lower uptake compared with the general population, where this has been measured.[2 9–14]

A current challenge is developing tailored vaccination interventions, services and systems which adequately respond to the needs of migrant populations.[15 16] Many governments did not include migrants well in their national plans at the start of the pandemic or adequately tailor health information to their linguistic needs and cultural preferences (eg, only 6% (3/47) of Council of Europe member states translated information on testing or healthcare entitlements into a foreign language).[6 17 18] In the UK, funding was mobilised to increase engagement with specific ethnic minority groups reporting lower levels of COVID-19 vaccine intent or uptake, including through outreach activities and the development of culturally relevant health information and messages.[19 20] However, few initiatives have specifically focused on understanding drivers of uptake in migrants (which is critical to increase uptake), actively involved migrant populations in the coproduction of vaccine interventions, or considered how rapid emergency outreach might erode trust with already disenfranchised groups who—until the pandemic—had felt largely ignored.[21–23]

Migrants are extremely heterogeneous and their reasons for undervaccination are variable, multiple and complex. Depending on their migration status and the influence of social determinants of health, these may include barriers before, during and after migration. Our recent systematic review[24] confirmed that access barriers, including language, literacy, communication, practical, legal and service barriers, are particularly important barriers to vaccination for migrants in transit and host countries, and that specific factors, including country of origin, having more recently migrated and being an asylum seeker or refugee, could be determinants of underimmunisation in migrants. Stigma, discrimination, xenophobia and racism are known to impact on access to health services in these populations.[1] Adult and adolescent migrants are also thought to be at risk of remaining undervaccinated for routine vaccinations after migration due to a lack of guidance (or implementation of guidance) on offering catch-up vaccinations and because, unlike children, they are not routinely incorporated into vaccination programmes on arrival in most European countries, including the UK.[25] Literature on COVID-19 vaccination barriers in migrants is more limited, but recent studies have pointed in particular to access barriers including language and communication issues, as well as lack of confidence stemming from mistrust of government and health authorities[26] and the influence of pervasive factors including structural racism, marginalisation and discrimination.[9 12 26–30]

WHO's new Immunization Agenda 2030[31] emphasises the need for ensuring equitable access to vaccination for all populations, and promotes integrating vaccination throughout the life course and catching up adolescent and adult migrants with missed vaccines, doses and boosters, including COVID-19 vaccines, to close immunisation gaps. The Regional Risk Communication and Community Engagement Interagency Working Group has also set out four strategic objectives for reducing the negative impacts of COVID-19, including that responses and strategies are community led, data driven, collaborative and reinforce capacity and local solutions.[32] Participatory approaches, including community-based participatory research (CBPR), promote these principles and emphasise inclusivity and power sharing in conducting research. They are likely to be more effective than traditional research approaches when working with underserved and marginalised individuals and populations who may have reason not to wish to trust or engage with institutions, because they lift perceived barriers to involvement.[33 34] Rather than doing research 'on' populations, participatory research actively involves those affected by the issue being studied as equal partners in the research process, so that research is done 'with' populations and value is given to the subjectivity of lived experience in the creation of knowledge. In this way, research is embedded within, conducted in collaboration with and tailored to a specific community or population.[35 36] The relevance of participatory approaches to migrant health

research and strengthening vaccination services has been noted[16 37] and evidence shows that interventions driven by insights from the communities they are designed to serve are more cost-effective and lead to better results for health behaviour outcomes than traditional interventions.[38 39] However, much research into migrant health is still driven largely by the interests of academics, policymakers and clinicians rather than by the communities directly affected by the issue being studied.[37] Implementation of policies promoting the inclusion of migrants in decision-making across countries is also inconsistent.[33] A global systematic review of studies that used participatory approaches with migrants found important shortfalls, with few studies actively including migrants in all research stages, and generally poor reporting of how participatory research approaches were used and principles upheld.[40]

Rather than addressing migrants as a single, homogeneous group or retrofitting public health initiatives originally designed for the general population, there is a need to actively involve specific migrant subpopulations in co-constructing knowledge about their lived experiences to inform the design of more sensitive health and vaccination initiatives which adequately respond to their needs, if we are to tackle existing health inequalities. For addressing COVID-19 vaccination inequities specifically, there is a need for more nuanced research into the drivers of COVID-19 vaccine uptake within and between migrant populations to advance understanding in this field and translate knowledge to practical action and interventions which account for migrants' unique cultural identity, beliefs and perspectives. While COVID-19 is the focus of this paper, similar gaps and opportunities exist with regard to routine vaccinations and in other disease areas, which require urgent focus.

The aim of this study is to use CBPR approaches to involve a specific subpopulation of migrants (in this case, adult Congolese migrants) in Hackney in the codesign of a tailored intervention to increase COVID-19 vaccine uptake. It seeks to (1) gather information about the local, sociocultural and historical context, (2) understand adult Congolese migrants' attitudes, beliefs and experiences relating to vaccination in general, COVID-19 vaccination and other lived experiences of UK healthcare and vaccination policies, (3) understand local pathways, processes and services, and considerations for implementation of interventions with professional stakeholders and (4) codesign a tailored intervention to strengthen COVID-19 vaccine uptake with Congolese migrants, which can be formally evaluated.

## METHODS AND ANALYSIS
### Context
Around 16 000 migrants from the Democratic Republic of Congo (DRC) are thought to live in the UK,[41] many of whom fled conflict and political instability and came to the UK to seek protection beginning in the late 1980s and 1990s[42] and continuing to the present

day.[43] Literature about Congolese diaspora in the UK is scarce. In December 2020, just before the UK government began rolling out the first COVID-19 vaccines, Congolese leaders of a small community-based organisation supporting Congolese migrants in London, UK (Hackney Congolese Women Support Group), voiced concerns during a community forum that misinformation about the COVID-19 pandemic and COVID-19 vaccines was spreading within their community, sparking widespread confusion and fear, with a large proportion of the community reluctant to get vaccinated. To the best of our knowledge, at this time there were no published data on Congolese migrants' vaccination attitudes, behaviours and beliefs in the UK.

### Forming a collaboration
Prior to study conception, exploratory workshops were held with representatives from various refugee and migrant populations in City & Hackney, London, UK. These were co-led by an academic researcher (AFC) and a community coordinator (CH) and facilitated by existing relationships and trust between CH and the local community organisations supported by the Hackney Refugee and Migrant Forum (within Hackney CVS). Three online meetings were held in December 2020 to February 2021 and refugee and migrant 'experts by experience' were invited to share their views and concerns regarding local unmet needs and discuss potential solutions and courses of action, with a particular focus on COVID-19. Hackney Congolese Women Support Group (LML, LMK, SN) was one of the local charities to attend and their participation led to further discussions about a potential research collaboration, particularly because of their small size, limited funding success to date and clear unmet needs in their community regarding the national COVID-19 response that they highlighted. The three organisations (St George's University of London, Hackney CVS and Hackney Congolese Women Support Group) discussed potential ways of working together to identify solutions starting with the needs and desires of the target population, before deciding to form a partnership or 'coalition' to codesign and lead the study. All partners discussed relative experiences, expectations, goals, timelines and budget, and used their respective assets to increase understanding of the other coalition members. For example, AFC and CH helped to create understanding of possible research approaches and methods, ethics, rights and ownership and empower the Congolese partners to participate with full voice; LML, LMK and SN advised on local preferences, customs and values that should be respected and provided valuable context on the lived experience of the target population.

### Study coalition and reflexivity
The coalition includes three women with lived experience as a Congolese migrant or descendant in the target population in London (LML, LMK, SN), one woman (CH) representing the local community and voluntary

sector and one woman (AFC) representing academia. Each of the coalition members holds positions of both privilege and marginality and the influence of these positions with respect to each other and the target population will be considered reflexively throughout the course of the study. Although there are some shared characteristics between all members of the coalition, AFC and CH generally consider themselves 'outsiders' and LML, LMK and SN consider themselves 'insiders' to the study population.

## Study planning

The coalition held three 2-hour planning meetings in November to January 2021 to agree to roles, responsibilities, study aims and objectives and plan the study (eg, recruitment, data collection, analysis, dissemination plans), with additional meetings to prepare and refine the study tools. Further meetings and reflection sessions are planned. AFC led one half-day training session for the coalition on qualitative interviewing techniques and one half-day session to practise, pilot test and refine the interview topic guide. CH provided additional training on facilitation skills. The coalition chose a CBPR approach, where partners are equals and actively involved in all stages of the research, and agreed to plan through an equity lens and prioritise building relationships and trust. The study was named 'Lisolo Malamu', meaning 'Good Talk' in Lingala (suggested by Hackney Congolese Women Support Group), to reflect the aim of facilitating dialogues and meaningful conversation around

COVID-19 vaccination and other health topics with their community.

## Study design

'Lisolo Malamu' ('Good Talk') is a CBPR study which uses coproduction and qualitative research methods to engage Congolese migrants in developing a tailored intervention to increase vaccine uptake. It involves three main activities: (1) community days, involving qualitative in-depth interviews (IDI) and interactive posters with Congolese migrants, (2) IDIs with local clinical, public health and community stakeholders and (3) codesign workshops with Congolese migrants. The principles of design thinking (an iterative, solutions-based approach to problem solving that starts with the needs and desires of the target population)[44] and behaviour change theory[45] will be used to support intervention codevelopment. An evaluation component will be embedded across all activities. The study process is illustrated in figure 1. Good practices, challenges and facilitators relating to the implementation of the study and the method of using codesign will also be documented.

The study was codesigned by an academic–community partnership ('coalition') which includes three members of the Congolese target population (described earlier). The coalition cowrote this protocol and will participate in all stages of the research and dissemination, including as peer researchers. This protocol reports on the decisions

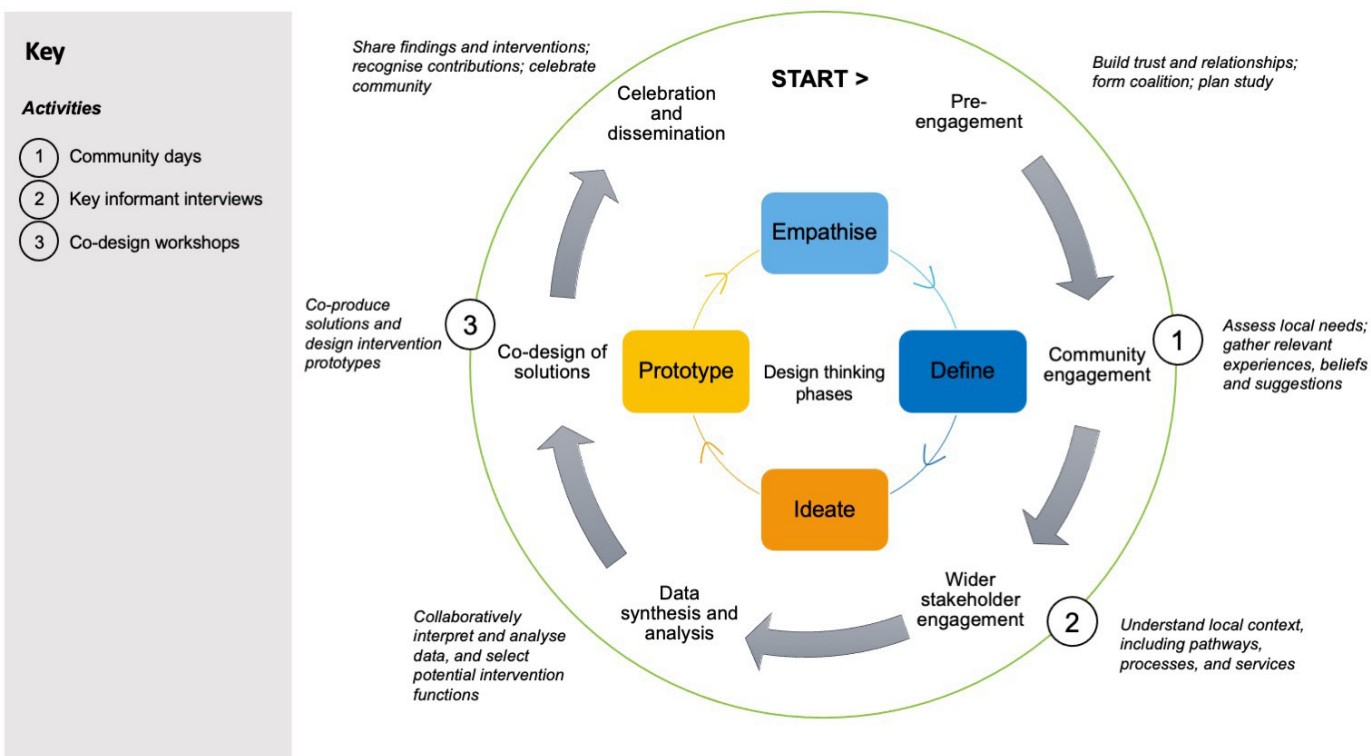

**Figure 1** Study process and activities mapped to the four design thinking phases: empathise (with target population); define (target population's needs, problems and insights); ideate (challenge assumptions and generate ideas for innovate solutions); prototype (start creating solutions).[44]

**Table 1** Inclusion and exclusion criteria

| Target population | Inclusion criteria | Exclusion criteria |
|---|---|---|
| Migrants | ▶ Born in the Democratic Republic of Congo (DRC).<br>▶ Aged 18 or above.<br>▶ Currently residing in the UK.<br>▶ Willing and able to give informed consent. | ▶ Not migrant as per earlier definition.<br>▶ Not born in DRC.<br>▶ Below the age of 18.<br>▶ Temporarily in the UK for holiday, visiting friends/family or other reasons.<br>▶ Individuals who may lack the capacity to consent, as determined by the mental capacity act framework. |
| Local stakeholders | ▶ Aged 18 or above.<br>▶ Volunteer or employee of a local group, organisation or business that has a vested interest in the health of the target community, such as local government, public health, National Health Service (NHS), community and faith-based organisations.<br>▶ Willing and able to give informed consent. | ▶ Not a local stakeholder as per earlier definition.<br>▶ Below the age of 18.<br>▶ Individuals who may lack the capacity to consent, as determined by the mental capacity act framework. |

made regarding the study design to date and the full process will be written up at the end of the study.

### Setting and population

The study is being carried out in Hackney, London, UK, a highly diverse London borough, in which more than 89 languages are spoken and around 40% of the population come from black and minority ethnic groups.[46] It was the 11th most deprived local authority in England in the Indices of Deprivation 2015.[47] The study will be conducted with adult migrants (>18 years) from the DRC and with local clinical, public health and community stakeholders based in Hackney. Specific inclusion and exclusion criteria are described in table 1. Hackney is thought to host one of the largest communities of Congolese migrants in the UK.[48]

### Recruitment

The study seeks to recruit approximately 30 Congolese migrants living in and around Hackney, London, UK to participate in semistructured qualitative interviews, 6–8 migrants to participate in the codesign workshops and approximately 4–6 local (to Hackney) professional stakeholders to participate in the key informant interviews. Hackney Congolese Women Support Group will lead the recruitment of local migrants using word of mouth, flyers codeveloped by the coalition and additional snowball sampling techniques. Professional stakeholders will be recruited through publicity among the coalition's networks (eg, email bulletins, word of mouth, local meetings, advertisements). Participants will be compensated according to National Institute for Health Research (NIHR) guidance[49] and reasonable expenses (travel, childcare, etc) will be paid.

### Data collection and activities

The study data and data collection methods are described in table 2. Due to cultural preferences, data will be collected face to face (COVID-19 restrictions permitting).

Translated participant information sheets will be distributed at least a week in advance of interviews, with participants given the chance to ask questions and decide whether they would like to participate. Written informed consent will be obtained in writing prior to starting the interview. Interviews with migrants will be conducted by four members of the coalition in Lingala, French or English, depending on the participant's preference (LML, LMK and SN are trilingual; AFC speaks English and will use an interpreter as required). Interviews with professional stakeholders will be conducted in English by AFC; codesign workshops will be cofacilitated by the coalition in Lingala, French and English. Qualitative interview data will be collected with a semistructured pilot-tested topic guide, which will be used flexibly. Interviewers will meet regularly during the data collection period to debrief on the interview process, discuss data and adapt the topic guide if required.

### Activity 1

'Community days' will be held, during which approximately 30 semistructured, in-depth, qualitative interviews with Congolese migrants will be conducted to explore beliefs, perceptions and experiences relating to routine and COVID-19 vaccination, UK healthcare and policies, and obtain suggestions for novel vaccination interventions. Additional data and insights about the local, socio-cultural and historical context and Congolese culture, customs and preferences will be collected through interactive posters in the social space. Post-interview evaluation forms and sociodemographic surveys will be collected. Information about local services (eg, educational classes, housing) will be available and referrals will be facilitated by CH. Community days will be held at a community centre near a local market attended by many Congolese for their weekly shopping, and planned to coincide with market days to encourage attendance. At the time of submitting this protocol, two community days have been

**Table 2** Study activities, data and data collection methods

| No | Activity | Population | Data generated/collected | Data collection methods | Person(s) responsible |
|----|----------|-----------|--------------------------|-------------------------|-----------------------|
| 1 | Community days (n~3) | Congolese migrants (n~30) living in and around Hackney, London, UK | ▶ Information about the local, sociocultural and historical context, customs and preferences.<br>▶ Beliefs and experiences related to routine and COVID-19 vaccination and other lived experiences of UK healthcare and vaccination policies.<br>▶ Suggestions for engagement approaches and interventions.<br>▶ Sociodemographic information. | ▶ IDIs (n~30).<br>▶ Post-it notes/interactive posters/graffiti walls.<br>▶ Sociodemographic surveys. | LML, LMK, SN and AFC will obtain informed consent and conduct IDIs. CH will manage logistics and registration, ensure participants are welcomed and comfortable and support linkage to wraparound services. |
| 2 | Key informant interviews | Local clinical, public health and community stakeholders (n~6) | ▶ Role and relationship with the Congolese community.<br>▶ Description of local pathways, processes and services.<br>▶ Suggestions for potential interventions and considerations for implementation. | ▶ IDIs. | AFC will obtain informed consent and conduct IDIs. |
| 3 | Codesign workshops (n~2) | Congolese migrants (n~8) | ▶ Codevelopment of and iteration on intervention prototypes. | ▶ Participatory workshops. | LML, LMK, SN, AFC and CH will facilitate workshops. |
| N/A | Evaluation | All populations plus community coalition | ▶ Feedback on involvement in codesign process.<br>▶ Feedback on participation in study activities (IDIs, workshops).<br>▶ Feedback on final prototype. | ▶ Evaluation forms/questionnaires.<br>▶ Voting. | CH will manage evaluation data with support from coalition. |

IDI, in-depth interview.

held, with interviews conducted in private rooms and a central social area provided for the community to gather over Congolese food and music.

### Activity 2

Approximately four to six in-depth, online interviews will be conducted with local key informants/professional stakeholders (eg, local general practitioners/nurses, clinical and public health staff, religious leaders and relevant community organisations in Hackney) to explore their role and relationship with the Congolese community, understand local pathways, processes and services and discuss potential interventions and considerations for implementation.

### Activity 3

Approximately two codesign workshops will be conducted in person with two groups of four to six Congolese migrants who participated in the IDIs (activity 1) to discuss and iterate on intervention functions and create an intervention prototype.

### Evaluation

Activities will be evaluated with feedback from participants and community feedback on the final intervention prototype will be sought at the celebration event.

### Data analysis and preparation of initial intervention prototypes

Qualitative interview and consensus workshop data will be analysed collaboratively by the coalition, to enhance understanding, interpretation and reflexivity,[50] manually and in NVivo software (Mac version). Anonymised digital recordings will be translated into English and transcribed verbatim by an independent professional translator, and transcripts, field notes, anonymous evaluation forms and other data collected during the activities (Post-it notes, posters) will be imported into NVivo for coding and analysis. Sociodemographic data will be entered into Excel, aggregated and summarised using descriptive statistics.

Analysis will take place in two stages, exploring both inductive and deductive orientations to data. The first stage will follow the six steps of reflexive thematic

analysis[51 52]: (1) data set familiarisation, (2) coding, (3) initial theme generation (whereby themes are patterns anchored by a shared idea, meaning or concept), (4) theme development and review, (5) theme refining, defining and naming and (6) writing up.[52] This stage will be more inductive. All members of the coalition will have access to the study data and will hold specific responsibilities to support the collaborative process: the academic researcher will manage the data and serve as a 'facilitator', guiding the coalition through the analytical steps to encourage and support their active participation. The community partners will facilitate member checking the study data with participants and triangulation of sources. Reflexive thematic analysis was chosen because it values subjectivity in knowledge creation, helps locate personal experiences within wider sociocultural contexts and is suited to research that needs to generate practical and actionable outcomes.[53]

The second stage will involve mapping the data to the theoretical domains framework (TDF)[54] and behaviour change wheel (BCW)[55] to identify behavioural components and potential intervention functions (defined as broad categories of means by which an intervention can change behaviour) needed to change behaviour.[45] The comprehensive and practical TDF and BCW[45] were selected to guide intervention development because they were specifically developed for implementation research, and support identifying changes at individual, organisational and system levels, and making policy recommendations. We expect this stage to be more deductive, with the analysis shaped by existing theoretical constructs. The compatibility of the two approaches will be critically discussed in the write-up. Candidate intervention functions will be selected by the coalition using the Affordability, Practicability, Effectiveness/cost-effectiveness, Acceptability, Side effects/safety, Equity criteria,[45] with approximately two suitable intervention functions taken forward to the codesign workshops with the Congolese study population. These intervention functions will be the starting point for the workshops, and potential intervention strategies involving these functions will be discussed, iterated on and tailored with the participation of the community, with the end goal being to coproduce a single, detailed intervention prototype. Any summary notes from the workshops and photographs of visual data generated (eg, Post-it notes, illustrations, etc) will subsequently be imported into NVivo software for data management and further analysis by the coalition.

## Schedule
The planned duration of the study is 12 months, starting from November 2021 and ending in November 2022.

## Support for partners
Study partners from Hackney Congolese Women Support Group and Hackney CVS will be financially compensated for their time and effort.[49] All study resources and expenses will be paid for by the project budget managed by the St George's migrant health research group. Non-financial contributions to Hackney Congolese Women Support Group include honorary library membership, training and upskilling opportunities and grant writing support.

## Patient and public involvement
Patient and public involvement is embedded throughout the participatory study design and approach. An independent patient and public involvement board (St George's Migrant Health Research Group NIHR Project Board) comprising five adult migrants with lived experience of accessing healthcare in the UK will also be consulted at significant points over the course of the study.

## ETHICS AND DISSEMINATION
This study was granted ethics approval by the St George's University Research Ethics Committee (REC reference: 2021.0128). A celebration event and webinar for participants, the local community and professional stakeholders will be organised at the end of the study to show impact and recognise contributions. The study findings will be disseminated at local, national and international levels, including through conferences, policy, stakeholder and voluntary/community sector meetings, peer-reviewed journals, a PhD thesis and multimedia outputs (eg, video clips and tweets). Research data and outputs will be stored in the St George's Research Data Repository. Recommendations for a future larger scale study and testing of prototyped interventions will be made.

**Author affiliations**
[1]Migrant Health Research Group, Institute for Infection and Immunity, St George's, University of London, London, UK
[2]Hackney Refugee and Migrant Forum, Hackney Council for Voluntary Service, London, UK
[3]Hackney Congolese Women Support Group, London, UK
[4]Doctors of the World UK, London, UK
[5]Institute for Infection and Immunity and Population Health Research Institute, St George's, University of London, London, UK
[6]Faculty of Health, Science, Social Care and Education, Centre for Applied Health and Social Care Research, Kingston University, Kingston, UK
[7]Our Future Health, Manchester, UK

**Acknowledgements** We thank the Hackney Congolese Women Support Group, Hackney Refugee and Migrant Forum and Hackney CVS, members of St George's Migrant Health Research Group NIHR Project Board and the community representatives and community organisations consulted during our pre-engagement work. We thank all the participants in our study and in particular the Congolese migrant communities in North East London.

**Contributors** AFC, SH, ASF, LML, SN, LMK and CH collectively discussed, conceived and codesigned the study and prepared all of the study documents (PIS, ICF, recruitment flyers, topic guides, surveys, etc). LML, SN and LMK recruited participants. AFC, LML, SN and LMK collected the data. AFC, LML, SN, LMK and CH collaboratively analysed and interpreted the data, with input and comments from SH, ASF, YC, FK, TV and LPG. AFC, LML, SN, LMK and CH organised and hosted the study celebration. AFC wrote the first draft, with input from the coalition (SH, ASF, LML, SN, LMK, CH). All authors (AFC, SH, ASF, LML, SN, LMK, CH, YC, FK, TV, LPG) reviewed and commented on a final version.

**Funding** This work was supported by the National Institute for Health Research (NIHR Advanced Fellowship 300072). SH and AFC are additionally funded by the Academy of Medical Sciences (SBF005\1111) and WHO. SH acknowledges funding from the Novo Nordisk Foundation/La Caixa Foundation (Mobility–Global Medicine and Health Research grant).

**Disclaimer** The funders did not have any direct role in the writing or decision to submit this manuscript for publication. The views expressed are those of the author(s) and not necessarily those of the NHS, the NIHR or the Department of Health and Social Care. The funder of the study had no role in study design, data collection, data analysis, data interpretation or writing of the report.

**Competing interests** None declared.

**Patient and public involvement** Patients and/or the public were involved in the design, or conduct, or reporting, or dissemination plans of this research. Refer to the Methods section for further details.

**Patient consent for publication** Not applicable.

**Provenance and peer review** Not commissioned; externally peer reviewed.

**ORCID iDs**
Alison F Crawshaw http://orcid.org/0000-0003-0450-7258
Lucy Pollyanna Goldsmith http://orcid.org/0000-0002-6934-1925
Sally Hargreaves http://orcid.org/0000-0003-2974-4348

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
