## [Reviewer comments · BMJ Open]

ARTICLE DETAILS

TITLE (PROVISIONAL)	Co-designing an intervention to strengthen COVID-19 vaccine uptake in Congolese migrants in the UK (LISOLO MALAMU): a participatory qualitative study protocol
AUTHORS	Crawshaw, Alison; Hickey, Caroline; Lutumba, Laura Muzinga; Kitoko, Lusau Mimi; Nkembi, Sarah; Knights, Felicity; Ciftci, Yusuf; Goldsmith, Lucy; Vandrevala, Tushna; Forster, Alice S; Hargreaves, Sally

VERSION 1 – REVIEW

REVIEWER	Maria Ganczak University of Zielona Gora, Department of Infectious Diseases
REVIEW RETURNED	02-Jul-2022

GENERAL COMMENTS	While I think this is an important topic that warrants investigation, there were several issues with both the study design, and the protocol itself that are significant enough that they undermine the contributions of the study protocol. I have a number of reservations about this protocol. They are outlined below. Firstly, in the Methods and analysis section the authors state “the study will run from approximately November 2021-November 2022”. The fundamental question is how to assess a study which is ongoing. In my opinion the revision should take place before the study begun. For this ongoing study (8 out of 12 months!), it is generally the case that no changes can be made to the methodology. As such, my revision is generally based on clarifications for the rationale and details relating to the methods. The authors further state that “COVID-19 vaccination provides a unique entry-point and opportunity to explore these issues in tandem with addressing routine immunization gaps and developing more culturally sensitive routine vaccination services”. COVID-19 vaccination among adult migrants requires different principals than catch-up vaccinations. The latter ones can be offered to adult refugees and migrants as they may have missed childhood vaccinations or the booster doses and could be at increased risk for VPDs during adulthood. The researchers should clearly define the study objective(s), specifically which vaccine it refers to as well as the outcome(s). An “intervention to strengthen vaccine uptake” seems quite enigmatic. The study population are adult migrants from the DRC and Republic of Congo, Angola or another Lingala-speaking region of Central Africa, living in the UK. This has not been addressed in the title of the study: “... to strengthen vaccine uptake in Congolese migrants”.
--

	What are the reasons for the discrepancy between the title and sampling? The main limitation of this study is that “co-designed intervention prototypes will not be formally implemented and evaluated in this study, however recommendations will be made so that this can be done in a future phase”. This can be done in a future phase, however, this might be the authors wishful thinking. Much better approach would be to design a study which works not only on an intervention prototype but also on an intervention implementation.
--	--

REVIEWER	Inge Smit University of Cape Town, Obs & Gynae
REVIEW RETURNED	31-Jul-2022

GENERAL COMMENTS	Your study addresses a fundamental issue concerning vaccine uptake and working with the targeted community to develop strategies is a much-needed approach with will help this population with accessing vaccines. The minor revision is that the informed consent is not detailed in the article. How will consent be obtained, and will it be available in the different languages?
---

REVIEWER	Dorota Chapko Imperial College London, School of Public Health
REVIEW RETURNED	19-Aug-2022

GENERAL COMMENTS	Review Thank you for providing me with an opportunity to review this protocol. I reviewed the protocol together with my colleague Kabelo Murray, Patient and Public Involvement Manager at the Applied Research Collaboration, Northwest London, Imperial College London. As a team, our expertise lies in participatory approaches in health research, life course brain health, mixed-methods data science and the interaction between one’s ethnicity, race, and gender with other identities and their impact on one’s health. At the start, we would like to emphasize the ‘participatory’ approach in co-designing the protocol which is uniquely placed in the space of research protocols and academic health research overall. This perhaps should not be extraordinary, but we would like to congratulate the entire team for bringing together academics, professionals, and experts by experience – a unique combination of team members which should be widely utilized in the space of academic health and social care research. Noting multiple strong elements in the research protocol, here we would like to offer suggestions for reflections and improvements, largely driven by our expertise in and passion for community-based participatory research as well as de-colonising practices within academic research and academia. General introductory comments: Of course, papers you may have reviewed may use these terms/general language but that does not license them. We need to speak about race, ethnicity, racism, decolonization, stigmas, stereotypes and populations in an honest and respectful way. Just because the literature reviewed for this study echoed these dated and inappropriate groupings of and manners of speaking about Blackness does not mean that it is okay. There is a surprisingly low
--

standard for racism within academia and we all have to actively do our part to shift this on a large scale. The comments below are not harsh or overly picky, they are honest and need to be seen as a starting point for how we speak about race and people. We cannot continue to group individual humans under large broad brushstrokes that erase their culture and historical differences.

Specific Comments:

Introduction:

Page 3 lines 8-18:

“Adult migrant” refers to an extremely large group of people that are, by no means, homogenous. This paper, however, focuses specifically on Black migrants from African countries. The use of “adult migrants” in reference to the actual group highlighted in this study implies that all adult migrants are Black and or of African decent – which we know is not true. Additionally, not everyone from Africa is Black, perhaps are more nuanced and appropriate definition of race and ethnicity and how this is defined (appropriately) is necessary. A differentiation between migrants and forced migrants is also necessary. This echoes larger conversations about how society classifies someone as a migrant versus an expat and how this classification often falls on racial and ethnic lines.

Furthermore, the inequities this paper refers to disproportionately impacts migrants of colour, specifically Black African migrants – this is an important distinction. Additionally, grouping all Black African migrants into this single category also echoes broader views of Africa that ignores the vast diversities across the continent. If this study refers to a specific group of people, this needs to be consistent because equating the experiences and behaviours of one group to an entire continent also echoes similar racialized generalizations of the African continent and those that originate from it. When referring to developing more “culturally-sensitive” routine vaccination services one must acknowledge that there are several very unique cultures present within the continent that no single culture can be used to group the entire population (which I believe is a common implication across this article).

It is clear that this paper focuses on Congolese migrants – this cannot be extrapolated across the continent or the entire grouping of “adult migrants” without subscribing to harmful stereotypes, generalizations and vague classifications of a massive and extremely diverse continent. This diversity echoes across the migrant populations within Europe, the UK, London and Hackney. Even within specific burrows one cannot generalize about the entire migrant, African migrant, Black migrant, Black or populations of colour. These are not homogenous groups and research needs to push back against this habit instead of perpetuating it, especially in the space of community-based participatory research.

Page 4 Lines 29-44

In the Limitations section this paper speaks to the ability to “...draw conclusions of other Black migrants who face similar historical and cultural barriers...” – in reference to the above comments on the harmful generalization of Black populations, this statement is perhaps inappropriately phrased if not a bit tone deaf. How, empirically, could the Congolese migrant experience be reflected on that of South African migrants? Or Nigerian - as another example? The cultural and historical barriers may be common in terms of colonization, racism or historical oppressions and contemporary experiences of racism but these aspects are not culture. By concentrating the commonalities of vast populations to historical injustices as opposed to (or instead of) recognizing the very real and historical uniqueness and diversities between these countries

restricts these identities to nothing but what has happened to them and not what these cultures actually are (see a comment on 'community assets' below).

Page 5

Although the reasons given for low-vaccination rates in low-and middle-income countries, this paper does not recognize the very recent history of colonization and the damage and destabilization this has done to many countries around the world. This perhaps is more important a factor to inequities in public health access across the world than, for example, "differing vaccination schedules". Perhaps discussions on high-income countries buying out vast stores of vaccinations ahead of poorer countries is also a significant reason for these disparities – which is also not a discussion around culture.

If one refers to reasons for undervaccination are "multiple and complex", it needs to be acknowledged that these complexities are largely as a result of historical and contemporary racism SPECIFICALLY amongst Black and Black African migrants. This is not to say that this is not an issue in other spaces however this paper focuses on a specific group but fails to acknowledge that this is not a coherent single identity. A more coherent definition of race and ethnicity is needed and more consistent terminology needs to be used. Not all migrants and people from low- to middle-income countries are Black or Black African.

Lines 24-33

Systematic racism, issues of trust and historical neglect from research and public health policy are not listed as reasons for undervaccination. This places responsibility on the person and not the system that has historically neglected and abused them. This is another shortcoming of research that further excludes populations of colour while enforcing harmful stereotypes without accepting responsibility of historical and current violence disproportionately felt by these communities.

Lines 51-60

The grouping together of African Americans, Black Africans and other Black populations is unacceptable. "Particularly Black groups" is a disrespectful grouping of non-homogenous and infinitely diverse populations of people that entirely removes their individuality as cultures, nationalities, identities, etc. Additionally, referring to the entire ethnic and racial group African American population as migrants is extremely inappropriate. Equating and reducing the very real atrocity of slavery to migration is not okay.

Page 6 Lines 11-47

Again, the inappropriate grouping of minorities and migrants. Does this paragraph refer to every migrant and minority experience? Does this mean to say that the experience of a Black Congolese migrant is grouped together with that of a Brazilian migrant, a British born LGBTQI+ identifying person and that of White South African? The imprecision of these statements means for broad generalizations that lean on assumptions that when the reader reads "migrant" or "minority" that they automatically think Black African.

Page 7 Lines 11-22

This language is inconsistent with previous and successive discussions. You now refer to people of Black ethnicity which is an entirely different but equally broad (and incorrect) grouping of diverse populations. Ethnicity and race are not the same thing. Blackness is a racial category, ethnicity is an expression or experience of culture and identity (Sneja 2007; Blakemore 2019; Bryce 2022). The term "Black ethnicity" is ill-fitting as a result of the previous differentiation.

Lines 25-57

This sections speaks to extremely different groupings of people as the same. These groups are migrants, refugees, asylum seekers, “those from Africa” and the Black community in Hackney (unsure what this means as the ‘Black community’ in Hackney does not host an individual identity other that this inappropriate classification. Not all migrants and refugees are Black and visa versa. Furthermore, “those from Africa” is another loose and inappropriate grouping. Compare these statements to referring to “those from Europe” or “those from Asia” and recognize how massive and vastly diverse those regions are. How can you empirically group a Zimbabwean to a Libyan as “those from Africa” (for example)? You can’t, these groups are extremely diverse and unique however, when viewed only under the lens of “these populations are Black and therefore the same” you are not able to refer to them as a single group. If the Congolese population is the fourth largest refugee nationality to settle in the UK, surely this deserves its own focus and grouping beyond Blackness? Additionally, the differences between refugee and migrant are important and at this section you acknowledge these populations in reference to refugee status and not as migrants – please keep this consistent while being aware of the dangers of ignoring the differences.

Overall, the data-driven justification for selecting adult Congolese migrants as the target population makes it somewhat confusing in this part of the protocol. What stands out, however, is the relationship built with community-based organizations during the pre-engagement work, perhaps with organizations representing Congolese migrants being more receptive and open to the idea of collaboration – it would be great if the authors could clarify this important piece of information around forming relationships within CBPR and not shy away from it throughout the document.

Also, it may be useful for the readers to make a distinction between qualitative research approaches versus participatory research approaches noting the difference/overlap, especially regarding the power dynamics. First, to our knowledge, there are very few qualitative studies that would involve migrants in the UK as informants (e.g. via qualitative interviews) as such interviews are usually performed with clinical and public health stakeholders. Second, moving one or even several levels up in the co-production ladder, and as the authors already point out, there are very limited initiatives to perform research along community partners and migrants with ‘lived experience’. It is also worth mentioning that different forms of CBPR approaches propose research ‘being led by’ a specific community, not merely ‘in collaboration with’ (e.g. ‘peer-research’, ‘inclusive research’ as different forms of CBPR).

The publication by Roura et al. 2021 that is already referenced in the protocol provides a great summary for the underlying principle within participatory research, here as applied to health research with migrants: “It aims to contribute to a shift from a deficit model that sees migrants as passively affected by policies to their reconceptualization as citizens who are engaged in the co-creation of solutions.” Perhaps the authors could emphasize that participatory approach is being used for this very reason, to move away from the deficit model and look into not only barriers and problems that the migrant communities are facing, but also their aspirations and potential suggestions for solutions.

Methods and Analysis:

Has the protocol been consulted or written together with an individual identifying as an adult Congolese migrant? If yes, this should be stated up front and the process explained.

	It is not clear how the behavioural change theory is going to be applied – as suggested by Michie et al. 2011, the BCW is a flexible and non-linear tool aiding the co-design of an intervention. However, whose behaviour are the authors going to change? The identification exercise is not clearly explained. Is it that by the current study design, it is going to be the change of behaviour among the adult Congolese migrants? Do the authors allow for the change to take place e.g. at the organizational level or among the different groups of stakeholders? In the methods, the authors explain that Activity No. 2 entails in depth interviews – earlier in the document, it is understood that only Activity No.1 involves in depth interviews (for example, in the abstract). In the abstract, the authors point out in the methods in Step 2 that “interviews and consensus workshops with clinical, public health and community stakeholders” will take place which is somewhat suggestive of the stakeholders ‘confirming’ or ‘debating’ the insights provided by the adult Congolese migrants via the interviews in Activity No.1. Only later in the text, towards the end of the introduction and in the methods section, it becomes apparent that the main objective of Step 2 is to “understand local pathways, processes and services” as an important component in the intervention co-design process. Perhaps this element could be introduced differently. In the methods section, we are still not very clear on the aims of the “consensus workshop” (emphasis: which seems to exclude people with lived experience of being an adult Congolese migrant). Please state clearly why this is the case. Page 8 Lines 1-47 “It seeks to i) gather information about and make sense of Congolese adult migrants’ beliefs” – this is an extremely inappropriate way to reference a foreign culture. Congolese culture does not need to be “made sense out of” as it is not nonsensical. You could refer to understanding, respecting, consolidating Congolese culture with UK policies but this idea of it needing to “make sense” (although, I am certain is unfortunate phrasing) is reminiscent of racist sentiments of African cultures being nonsensical. Of course, this statement in itself is not racist however, the use of it in this very specific and very historically charged context makes it inappropriate. This paper then begins to refer to Congolese migrants, again. To reiterate, migrants and refugees are not the same. Migration implies (and entails) the ability to return back to one’s home country whereas refugee refers to one who cannot return safely home. This distinction is extremely important in legislation and immigration policy and should be important in this study. Page 8, line 41: What does the evaluation component across all the activities entail? Setting and population: in the spirit of co-designed research and moving away from the deficit model, could the authors mention key elements of the communities’ assets? This may also relate to the BCW opportunity element. Making sense of Table 1: The grouping of Congolese and/or Angolan seems arbitrary and inappropriate without proper justification or explanation. Why not group Spain and France together as they both host the Basque people? Because we recognize them as nations and identities independent to each other. We strongly recommend providing readers with brief geographical and historical background to Congolese migration patterns, also within Africa, and Congolese identity alongside associated
--	---

	challenges and how it relates to this study. In short, how did the authors decide on the inclusion criteria re: specified countries and regions? Line 54: If you refer to someone as a “White migrant woman” then you have to refer to those who are Black migrants or you are just leaning into the assumption that all migrants are Black and therefore you only need to specify race when they are not Black. Study Team and Coalition and the notion of ‘power’: until this point, it has not been apparent that the coalition includes individuals identifying themselves as Congolese migrants which, of course, is the key in participatory research. I recommend making it clear at the outset. In general, throughout the research process, how decisions are made? What types of knowledge and experiences are prioritized and valued the most (e.g. academic vs. lived experience)? How about the notion of ‘power’ that is central to community-based research? How has it been negotiated and shared across the team members? What are the plans for sharing power in the long term with regard to all the Activities 1-4? Thank you for detailing the financial compensation and the non-financial contributions given that appropriate, fair and efficient budgeting is among the perennial challenges for PPIE and community-based participatory research overall. Could the authors provide the interview topic guide for review? Data collection: it is not clear how the authors will make sense of the different and the vast amounts of data throughout the study. Please clarify. Also, if possible, Table 2 should include another column (or otherwise) indicating who performs given activity. Reflexivity, which is central to high quality qualitative research practice as recognized by the authors, is poorly explained and operationalized throughout. How are the team members going to share their experiences and learnings with one another? What about the notion of trust? How has it been established? How are the elements of power, trust, and the practice of reflexivity going to be practiced and negotiated? Evaluation and feedback: what protocols are going to be utilized or co-designed? Are participatory approaches considered for the evaluation and feedback? Data analysis: We would strongly recommend the authors familiarize themselves with Braun and Clarke (2019) for a ‘fresher’ and re-visited perspective on thematic analysis by Braun and Clarke which, in our opinion, is relevant and suitable for participatory research practice. Based on this work, we would also encourage the authors to revisit the plan of deciding on the sample size based on saturation within the participatory paradigm. It is good to see that the socio-demographic data will be aggregated which is appropriate for the topic under investigation and for community-based participatory research. I suggest the authors provide a rationale for this. I understand that the qualitative and the quantitative data will be led by SGUL researchers ‘in consultation’ with the coalition and only the academic researchers will have full access to all the forms of data. Please provide a rationale for this. Since the qualitative interviews are going to be performed by 4 members of the coalition, it seems like a missed opportunity not to involve all the members in the analysis process. May I point the authors to a recent resource developed by the Patient Experience Research Centre at Imperial College London summarizing how to involve people with lived experience in qualitative data analysis: Peer Research Training Resource Faculty of Medicine Imperial College London Ethics and dissemination: given that non-academic partners will be
--	--

	conducting interviews, what are some of the anticipated ethical and GDPR-related implications of such practice? How are the authors going to identify and mitigate risks and opportunities involved? This paper hosted no discussion regarding safeguarding while identifying that they would be working with refugees and forced migrants. Do the authors have proper safe-guarding procedures in place for both researchers and study participants? For how long is the data going to be stored? How will the migrants be acknowledged across all the outputs? References: Blakemore, E. 2019. "Race and ethnicity: how are they different? National Geographic. Accessed from: https://www.nationalgeographic.com/culture/article/race-ethnicity Bryce, E. 2022. "What's the difference between race and ethnicity?" LiveScience. Accessed from: https://www.livescience.com/difference-between-race-ethnicity.html Sneja, G. 1997. "Postcolonialism and Multiculturalism: Between Race and Ethnicity." The Yearbook of English Studies 27: 22–39. https://doi.org/10.2307/3509130. Virginia Braun & Victoria Clarke (2019) Reflecting on reflexive thematic analysis, Qualitative Research in Sport, Exercise and Health, 11:4, 589-597, DOI: 10.1080/2159676X.2019.1628806
--	--

VERSION 1 – AUTHOR RESPONSE

Reviewer: 1

Dr. Maria Ganczak, University of Zielona Gora

Comments to the Author:

While I think this is an important topic that warrants investigation, there were several issues with both the study design, and the protocol itself that are significant enough that they undermine the contributions of the study protocol. I have a number of reservations about this protocol. They are outlined below.

Firstly, in the Methods and analysis section the authors state “the study will run from approximately November 2021-November 2022”. The fundamental question is how to assess a study which is ongoing. In my opinion the revision should take place before the study begun. For this ongoing study (8 out of 12 months!), it is generally the case that no changes can be made to the methodology. As such, my revision is generally based on clarifications for the rationale and details relating to the methods.

Many thanks for your detailed and helpful comments, which are much appreciated. To address your first point, this protocol was written prior to beginning the study (as part of the ethics application) and was submitted to the journal as a manuscript in March 2022. At this stage, our main activities had involved building trust and relationships between study partners and planning the study, and we had only just begun collecting data for activity 1. The stage of the study at the time of submission was made clear to the Editor who confirmed it met the journal’s criteria for submission. We understand the journal had difficulty finding appropriate reviewers, which may have contributed to you receiving it at this late stage, which is unfortunate. It is of course hard to revise a protocol now many months later into the project timeline, but we have tried to do this with integrity in response to reviewers’ comments, while keeping to the original study concept and design.

The authors further state that “COVID-19 vaccination provides a unique entry-point and opportunity to explore these issues in tandem with addressing routine immunization gaps and developing more culturally sensitive routine vaccination services”. COVID-19 vaccination among adult migrants requires different principals than catch-up vaccinations. The latter ones can be offered to adult refugees and migrants as they may have missed childhood vaccinations or the booster doses and could be at increased risk for VPDs during adulthood.

Thank you. We agree that different approaches are likely needed for broadly different types of vaccines and that the target populations may also differ between COVID-19 and catch-up vaccinations, however perhaps not exclusively. Evidence shows that many types of migrant and some specific migrant populations are under-immunised for routine immunisations, some of whom have also faced barriers to accessing and receiving COVID-19 vaccinations. Some of these barriers overlap some of the time and there is a need to think about innovative approaches to reach migrant populations, including through more inclusive policies and bundling of services.

We want to clarify that we did not mean to suggest that all the approaches required for delivering COVID-19 vaccines and catch-up vaccines are the same. We wanted to highlight the unique opportunity that COVID-19 vaccination has provided for thought and action around reaching underserved migrant populations and addressing immunisation gaps through novel and more community-centred approaches, thanks to the increased public interest and political will (particularly around engaging minoritised populations through public health messaging and vaccination campaigns) since the start of the pandemic.

The researchers should clearly define the study objective(s), specifically which vaccine it refers to as well as the outcome(s). An “intervention to strengthen vaccine uptake” seems quite enigmatic.

Thank you. While ‘an intervention to strengthen vaccine uptake’ is perhaps too vague from a traditional research perspective, given the participatory approaches used in this study, which aim to ensure the research topic is driven by the desires and needs of the target population, we felt we could not pre-define the intervention more specifically before the study had begun. For balance, we have amended the intervention description to ‘COVID-19 vaccine uptake’, as our earlier scoping workshops with community representatives indicated that this was their primary concern and area on which they would like to focus (please note, however, that the study did explore routine and catch-up vaccinations as well). We have not been able to expand on this in as much detail in the text due to word limits, however it will be covered further in the results paper when published.

The study population are adult migrants from the DRC and Republic of Congo, Angola or another Lingala-speaking region of Central Africa, living in the UK. This has not been addressed in the title of the study: “.... to strengthen vaccine uptake in Congolese migrants”. What are the reasons for the discrepancy between the title and sampling?

Thanks for raising this point. The discrepancy came about because our community partner organisation (Hackney Congolese Women Support Group) noted that although most of the community they support originates from DR Congo, some members come from the Republic of Congo, some from Angola, and others from other parts of Central Africa, but many share the common language of

Lingala, a language associated with a specific part of Central Africa which crosses political/geographical borders. For simplicity and because this is a study protocol, we have amended the inclusion criteria to include just those born in DR Congo, however in the results manuscript we will discuss the actual sampling that took place in the study (reflecting expansion of the study inclusion criteria if needed).

The main limitation of this study is that “co-designed intervention prototypes will not be formally implemented and evaluated in this study, however recommendations will be made so that this can be done in a future phase”. This can be done in a future phase, however, this might be the authors wishful thinking. Much better approach would be to design a study which works not only on an intervention prototype but also on an intervention implementation.

Thank you for this valid point. Unfortunately, long-term implementation and evaluation of the interventions is beyond the scope of the study’s budget. However, the participatory approach supports capacity building and skills development of the partner community-based organisation to continue the work and will actively seek out funding, commissioning and further partnership opportunities from local stakeholders throughout the course of the study. These and the other benefits/impacts of the study will all be reported in the final study write up.

Reviewer: 2

Ms. Inge Smit, University of Cape Town

Comments to the Author:

Your study addresses a fundamental issue concerning vaccine uptake and working with the targeted community to develop strategies is a much-needed approach with will help this population with accessing vaccines.

Many thanks for this positive review.

The minor revision is that the informed consent is not detailed in the article.
How will consent be obtained, and will it be available in the different languages?

Thank you for highlighting this oversight in the protocol - we originally only briefly mentioned the consent process in Table 1 due to challenges with the word count. As per ethics requirements, written informed consent was obtained from all participants prior to participating in the study. Participants were informed about the study at least 1 week in advance, in their language of choice. They were given a translated participant information sheet, told about the study, and given time to reflect and ask questions before deciding whether to participate. Written informed consent was taken on the day of the interview, immediately before the interview. We have now added a statement to the methods section to make this clear (p11).

Reviewer: 3

Dr. Dorota Chapko, Imperial College London

Comments to the Author:

Please refer to the attached file for all the comments

Thank you very much for your comprehensive comments on our manuscript. We have reflected on your points, particularly those about the language used. It is very helpful to hear your perspectives on some of the phrasings and language we commonly use and how these might be perceived as problematic or disrespectful, which was obviously not our intention. We take your points on board and have tried to reflect these better in the text. In particular, we have revised the introduction and added a text box to elaborate on some of the limitations of the language and existing data and how these can be particularly incongruent with a participatory, community-centred approach.

We were motivated to do this kind of specific, nuanced, and community-led piece of research which values individuals' lived experience for many of the very reasons you have pointed out, and with this in mind we will be more critical of the future literature/language we read and use as a result of your helpful comments.

As this is a medical journal, aimed at clinical and public health professionals, we felt it necessary to keep the original structure of the introduction, moving from big and broad (migrants internationally, big picture, gaps in current research and approaches) to a more specific focus on the precise aims of the study at the end. The results write up, which will be pitched to a social sciences journal, will focus much more on the specific Congolese experience and the relevant themes generated from the data.

We have also addressed your recommendation to not "shy away" from explaining the participatory and relationship-building steps in more detail and have added new sections after the introduction including: context, forming a collaboration, study coalition and reflexivity, and study planning. We agree it is important to shed light on these approaches (previously we tried to conform to a more traditional manuscript layout and word limits) and hope they are helpful to others working in this area or wishing to explore using these approaches.

We have clarified in the methods and analysis sections that the protocol was co-written by the coalition including those with lived experience as Congolese migrants. Hopefully this collaboration is now much clearer! The reference to behaviour change theory has been amended to clarify that this will address changes at multiple socio-ecological levels and make wider policy recommendations, therefore it will not solely focus on changing the behaviour of Congolese migrants but will seek to address structural and organisational factors; this will be elaborated on further in the write up. Study activities have been clarified, and consensus workshops removed as these were in fact removed from the design after submitting the protocol. The other points have been addressed in response to other reviewers and/or in the revisions in the text, particularly around reflexivity and reflexive thematic analysis. It would not be appropriate to share the topic guide for review as data has at this point already been collected, however this will be made available through a repository when the final results are published. All of the ethics, safeguarding and data storage queries were fully addressed during the ethics application, and ethics approval was granted in January 2021 before this protocol was submitted. Thank you very much for your comments.

VERSION 2 – REVIEW

REVIEWER	Inge Smit University of Cape Town, Obs & Gynae
REVIEW RETURNED	17-Oct-2022

GENERAL COMMENTS	The article reads easier and all comments or concerns by the reviewers has been addressed. Can't wait for the articles that will follow from this study and the interesting views the participants had and interventions that can develop after getting the communities feedback. A comment in respect to numbers used in the article numbers under 10 are usually written out in full but it your decision if it will help or hinder readability of the article.
--

REVIEWER	Dorota Chapko Imperial College London, School of Public Health
REVIEW RETURNED	22-Nov-2022

GENERAL COMMENTS	Once again, thank you for providing me with an opportunity to review the resubmission of the protocol. I reviewed it together with my colleague Kabelo Murray, Patient and Public Involvement Manager at the Applied Research Collaboration, Northwest London, Imperial College London. Thank you to the authors for a very detailed and kind response to our suggestions, clarifying queries, and points of concern. We believe that most of the suggestions have been taken on board and incorporated throughout the manuscript. We would like to note that the protocol in its current form is not perfect with respect to the presentation and representation of foreign-born individuals and migrants, specifically Congolese migrants, in the space of migration studies. The Box 1 in particular is a tiny bit peculiar but we think is an admirable response to the issues we had with language across the paper. As a field, we will need to do more, learn better and quicker to represent the populations, in this case Congolese migrants, and to appropriately reflect their needs and aspirations, in a way that is meaningful to them, with implication for how to perform research co-production with migrant populations. We also strongly believe that the research project, as outlined in the protocol, will provide a great basis for these required future learnings and we wish the research team all the best for the next steps in this endeavour.
--